# Enhanced Electrochemical Behavior of Peanut-Shell Activated Carbon/Molybdenum Oxide/Molybdenum Carbide Ternary Composites

**DOI:** 10.3390/nano11041056

**Published:** 2021-04-20

**Authors:** Ndeye F. Sylla, Samba Sarr, Ndeye M. Ndiaye, Bridget K. Mutuma, Astou Seck, Balla D. Ngom, Mohamed Chaker, Ncholu Manyala

**Affiliations:** 1Department of Physics, Institute of Applied Materials, SARChI Chair in Carbon Technology and Materials, University of Pretoria, Pretoria 0028, South Africa; ntoufasylla@gmail.com (N.F.S.); ssarr3112@gmail.com (S.S.); bridgetmutuma@gmail.com (B.K.M.); 2Laboratoire de Photonique Quantique, d’Energie et de Nano-Fabrication, Faculté des Sciences et Techniques, Université Cheikh Anta Diop de Dakar (UCAD), Dakar-Fann Dakar B.P. 5005, Senegal; nmaty.ndiaye@gmail.com (N.M.N.); balla.ngom@ucad.edu.sn (B.D.N.); 3Institut National de la Recherche Scientifique Centre—Énergie Matériaux Télécommunications 1650, Boulevard Lionel Boulet, Varennes, QC J3X 1S2, Canada; astou.seck@emt.inrs.ca (A.S.); chaker@emt.inrs.ca (M.C.)

**Keywords:** porous carbon, ternary composite, molybdenum oxide, molybdenum carbide, energy storage

## Abstract

Biomass-waste activated carbon/molybdenum oxide/molybdenum carbide ternary composites are prepared using a facile in-situ pyrolysis process in argon ambient with varying mass ratios of ammonium molybdate tetrahydrate to porous peanut shell activated carbon (PAC). The formation of MoO_2_ and Mo_2_C nanostructures embedded in the porous carbon framework is confirmed by extensive structural characterization and elemental mapping analysis. The best composite when used as electrodes in a symmetric supercapacitor (PAC/MoO_2_/Mo_2_C-1//PAC/MoO_2_/Mo_2_C-1) exhibited a good cell capacitance of 115 F g^−1^ with an associated high specific energy of 51.8 W h kg^−1^, as well as a specific power of 0.9 kW kg^−1^ at a cell voltage of 1.8 V at 1 A g^−1^. Increasing the specific current to 20 A g^−1^ still showcased a device capable of delivering up to 30 W h kg^−1^ specific energy and 18 kW kg^−1^ of specific power. Additionally, with a great cycling stability, a 99.8% coulombic efficiency and capacitance retention of ~83% were recorded for over 25,000 galvanostatic charge-discharge cycles at 10 A g^−1^. The voltage holding test after a 160 h floating time resulted in increase of the specific capacitance from 74.7 to 90 F g^−1^ at 10 A g^−1^ for this storage device. The remarkable electrochemical performance is based on the synergistic effect of metal oxide/metal carbide (MoO_2_/Mo_2_C) with the interconnected porous carbon. The PAC/MoO_2_/Mo_2_C ternary composites highlight promising Mo-based electrode materials suitable for high-performance energy storage. Explicitly, this work also demonstrates a simple and sustainable approach to enhance the electrochemical performance of porous carbon materials.

## 1. Introduction

The high demand for energy in conjunction with the rapid depletion of fossil fuels has made it essential to develop alternative energy sources. Various researchers have shown an increased interest in the development of clean, sustainable and renewable energy sources such as solar, wind and geothermal [1,2]. However, there are still challenges linked to the production and continuous supply of energy in large quantities from these renewable energy systems [3]. Therefore, in order to supply energy needs on a long-term basis, in addition to it being affordable, sustainable and environmentally friendly, it is important to find diverse, efficient, safe and flexible methods for its simultaneous generation and storage [4,5,6]. Batteries and supercapacitors are the most renowned energy storage devices with supercapacitors being given gross attention by energy researchers globally due to their high performance features such as high specific power, long life cycle and quick charge-discharge dynamics [7,8]. 

Supercapacitors are typically categorized into electrical double layer-capacitor (EDLC) and pseudo capacitors. The former operates on a charge storage process that relies on an electrostatic charge accumulation formed at the interfacial region of electrode/electrolyte while the latter depends on reversible faradic-type redox reactions at the electrode surface material [9,10]. 

Porous carbon such as carbon nanotube, graphene, activated carbon (AC) and carbon onions owing to their large surface area, good electrical conductivity and great stability are mainly investigated as electrode materials for EDLC [11,12]. Porous AC materials obtained from biomass waste (peanut shell, walnut shell, pinecone and so on) have gained interest due to their distinctive features of well-developed surface area with hierarchical pore structure, abundant availability of precursors sources, environmental friendliness and low costs [13,14]. In addition, surface functional groups obtained from the AC can favor an easy ion adsorption/desorption at the electrode/electrolyte interface leading to an optimum electrochemical performance [15,16]. However, there is still a need to improve their performance to meet the high-energy demand. 

To date, various methods have been reported to enhance the electrochemical performance of the porous ACs. For instance, the introduction of oxygen functional groups (ketone, ether, carboxylic acid, quinone, and so on) by oxidizing the porous carbon surface could promote the hydrophilicity and also surface reactivity of the carbon material [17,18,19]. Moreover, the presence of the surface oxygen-containing species not only provides some pseudo capacitance effect but also enriches the surface wetting capability which contributes a significant improvement in the capacitance and the overall specific energy/power of the carbon electrode material [19,20]. 

Song et al. [19] prepared O/S dual-modified nanoporous carbon (OSC) by a hydrothermal oxidation method using H_2_O_2_ (O) and H_2_SO_4_ (S) as oxidants. Their study revealed that the OSC electrode delivered a specific capacitance of 168 F g^−1^, 3.5 times higher than the pristine nanoporous carbon in 6 M KOH aqueous electrolyte due to the introduction of the oxygen functional groups on the surface of nanoporous carbon [19]. Another promising method is the heteroatom-doping (nitrogen, sulphur, phosphorous, boron and so on) of a porous AC matrix, which enhances the capability to store charge in the material. Therefore, heteroatom doping into the porous carbon framework results in the modification of the electronic structure, which can remarkably improve the electrical conductivity, the surface wettability, properties of the electrons donor and hence the electrochemical features of the porous carbon materials [21,22,23,24]. For instance, we reported previously in our study on the synthesis of nitrogen-doped peanut shell activated carbon (NPAC) by chemical activation and nitrogen-post-doping processes with KOH and melamine respectively. The NPAC showed a considerable increase in the specific capacitance value (167 F g^−1^ to 216 F g^−1^) for the non-doped PAC and NPAC electrodes in a 2.5 M KNO_3_ aqueous electrolyte [24].

Incorporating transition metal oxides/hydroxides or conducting polymer into the porous carbon network is also an effective strategy that can improve the electrochemical properties of these carbon materials. The integration of porous carbon with transition metal oxides (MnO_2_, NiO, Fe_3_O_4_, MoO_2_, etc.) forming composites synergistically combines the advantages and mitigates the limitations of both materials [25,26,27]. MoO_2_ is one of the most promising pseudocapacitive transition metal oxides which possesses several oxidation states (+2 to +6), high theoretical specific capacitance, excellent redox reaction capability, low electrical resistivity, good electrochemical activity and low cost [28,29]. 

Thus, the MoO_2_ incorporated into the porous carbon may provide more electrochemical active sites that could influence additional surface pseudocapacitive effect endowing enhanced electrochemical performance of the composite [30,31,32]. Lina et al. [30] synthesized MoO_2_ nanoparticles decorated into 3D porous graphene (MoO_2_-rGO) using a hydrothermal route. Their work indicated that the MoO_2_-rGO composite enhanced specific capacitance to 356 F g^−1^ as compared to the 3D porous graphene (244 F g^−1^) in 6 M KOH aqueous electrolyte [30]. 

Other studies have also shown the incorporation of transition metal carbide (Mo_2_C, W_2_C, TiC, etc.) in the carbon framework could improve the performance metrics of the electrochemical capacitors [33,34,35]. Among the transition metal carbides, Mo_2_C has recently attracted great interest due to the high specific conductance (1.02 × 10^2^ S cm^−1^), great conductivity (electrical and thermal) and good chemical stability [36,37,38]. Furthermore, these remarkable proprieties can offer additional actives sites, facilitate electron and ion transportation, good cyclic stability properties and a reduction of the charge resistance which could produce a relatively high electrochemical performance [38,39]. Hussain and co-workers [40] have prepared carbon nanotubes (CNTs) decorated with molybdenum carbide nanosheets (Mo_2_C@CNT) by a chemical reduction approach followed by carbonization. In their report, the hybrid composite Mo_2_C@CNT exhibited a specific capacitance value of 365 F g^−1^ which is 3.5 times higher than the CNTs (103 F g^−1^) in KOH electrolyte owing to the synergy between the Mo_2_C and CNTs [40].

Forming a ternary composite of metal oxide (MoO_2_), metal carbide (Mo_2_C) and porous carbon material result in a combination of the advantages of each component that could considerably enhance the electrochemical features of the energy storage device [38,41,42]. For instance, Ihsan et al. [36] have prepared a MoO_2_/Mo_2_C/C spheres by a two-step, hydrothermal process followed by a calcination procedure. 

Yang et al. [42] have also synthesized MoO_2_/Mo_2_C/C hybrid microspheres by a template-free method. Both studies showed a great rate capability, cycling stability, good capacity as anode materials for Li-ion batteries. However, to the best of our knowledge no reports exist on the application of these Mo-based materials/porous carbon (MoO_2_/Mo_2_C/C) ternary composites as supercapacitor electrodes. 

In this study, we have established a facile and low-cost approach of synthesizing a ternary composite (PAC/MoO_2_/Mo_2_C) for the first time as electrode material for supercapacitor by one-step pyrolysis route through varying mass ratios of ammonium molybdate tetrahydrate to porous peanut shell activated carbon (PAC) (1:0.5; 1:1; 1:2). The in-situ formation of MoO_2_ and Mo_2_C nanostructures incorporated into the PAC network were confirmed by the XRD, Raman, HRTEM, SAED, SEM, EDX mapping, as well as the XPS analysis. The obtained ternary composites portrayed interesting merits based on the existing incorporated nanostructures of the MoO_2_-Mo_2_C within the nanoporous PAC-based material including: (i) high specific surface area with hierarchically porous structure of the PAC, (ii) pseudocapacitive effect by the redox reaction of MoO_2_ and (iii) superior electrical conductivity and stability of the Mo_2_C. The best ternary composite (PAC/MoO_2_/Mo_2_C-1) exhibited superior capacitive performance in both half and full-cell test owing to the synergistic effect of the MoO_2_ and Mo_2_C nanostructures embedded into the PAC matrix. Moreover, our study demonstrates a simple and sustainable approach to enhance the electrochemical performance of porous carbon materials. 

## 2. Experimental 

### 2.1. Materials

In this study, all chemical reagents were used as obtained without any further purification. Ammonium molybdate tetrahydrate (NH_4_)_6_Mo_7_O_24_·4H_2_O, 99.98%), potassium nitrate, (KNO_3_, 99.99%), potassium hydroxide (KOH, 99%), polyvinylidene fluoride (PVDF, 99%), carbon acetylene black (CAB, 99.95%), hydrochloric acid (HCl, 37%) and N-methyl-2-pyrrolidone (NMP, 99%) were supplied from Merck (Johannesburg, South Africa). Argon gas (Ar, 99.99%) was purchased from Afrox (Johannesburg, South Africa) and polycrystalline nickel foam mesh (with 1.6 mm thickness, 420 g m^−2^ areal density) was obtained from Alantum (Munich, Germany).

### 2.2. Synthesis of the Peanut Shell Waste Derived Activated Carbon (PAC)

Peanut shell waste derived activated carbon (PAC) was prepared by a one-step chemical activation following a reported procedure from our previous study [24]. Briefly, the peanut shell waste raw material was mixed with KOH pellets in an optimized ratio by mass and then subjected to chemical activation at an elevated 850 °C temperature for 1 h to obtain the final product. 

### 2.3. Synthesis of the Peanut Shell Derived Activated Carbon/Molybdenum Oxide/Molybdenum Carbide (PAC/MoO_2_/Mo_2_C) Ternary Composites

The synthesized PAC sample was mixed with ammonium molybdate tetrahydrate in mass ratios of 1:0.5, 1:1 and 1:2 for the PAC to the inorganic salt in an agate mortar. Few drops of deionized water were added to the as-prepared mixture to ensure thorough mixing of both materials which was then loaded onto a porcelain boat and air dried at 80 °C for 12 h in an electric oven. 

The porcelain boat was transferred into a horizontal tube furnace and heated to 850 °C at a ramping rate of 5 °C min^−1^. The furnace was kept constant at this temperature for 1 h under 250 sccm of argon gas flow. After cooling down to room temperature, the ternary composites were obtained and labelled PAC/MoO_2_/Mo_2_C-0.5, PAC/MoO_2_/Mo_2_C-1 and PAC/MoO_2_/Mo_2_C-2 corresponding to the mass ratio of 1:0.5, 1:1 and 1:2, of PAC to the inorganic salt, respectively. The schematic procedure for preparing the PAC/MoO_2_/Mo_2_C ternary composites is illustrated in Figure 1. 

### 2.4. Physical Characterization

Powder X-ray diffraction (XRD) analysis of the PAC/MoO_2_/Mo_2_C samples was determined using a Brucker D8 Advance diffractometer using Cu Kα (λ = 1.5406 Å) radiation operating in the 2θ range of 10–80°. Raman spectra of the ternary composite materials were characterized on WITec alpha300 RAS+ confocal Raman microscope (WITec, Ulm, Germany) using a 532 nm excitation laser at a power of 5 mW.

High-resolution transmission electron microscope (HRTEM) micrographs and selected area electron diffraction (SAED) patterns were obtained using a JEOL JEM-2100F field emission gun transmission electron microscope (FEG-TEM) operating at 200 kV. Scanning electron microscope (SEM) micrographs and energy dispersive X-ray (EDX) mappings were carried out using a Zeiss Ultra-plus 55 field emission scanning electron microscope (FE-SEM). The SEM images and EDX mapping images were operated at 1.0 and 10 kV accelerating voltage respectively. The surface area distribution was performed by Brunauer-Emmett-Teller (BET) and porosity pore size by Barrett-Joyner-Halenda (BJH) methods on the Micrometrics TriStar II 3020 (version 2.00) system in a relative pressure (P/P_0_) range of 0.01–1.0 at 77 K. X-ray photoelectron spectroscopy (XPS) of the samples was obtained by a VG Escalab 220i-XL instrument equipped with a monochromatic Al-Kα (1486.6 eV) source of radiation.

### 2.5. Electrochemical Characterization

The PAC/MoO_2_/Mo_2_C electrodes were prepared by mixing the active material, carbon acetylene black (CAB) as conductive additive and polyvinylidene difluoride (PVDF) as binder in a weight ratio of 8:1:1, respectively in an agate mortar. Few drops of N-methylpyrrolidone (NMP) as solvent was added to the mixture to obtain slurry which was uniformly coated on nickel foam (NF) as current collector followed by drying at 80 °C for a period of 12 h in an electric oven.

The cyclic voltammetry (CV), galvanostatic charge-discharge (GCD) and electrochemical impedance spectroscopy (EIS) were investigated using VMP-300 16-channel potentiostat (Bio-Logic, Knoxville, USA) associated with EC-Lab^®^ (V11.33) software. The three electrode (half-cell) test was performed using the as-prepared electrode as working electrode, the glassy carbon as counter electrode (CE) and Ag/AgCl (in saturated 3M KCl) as reference electrode (RE). 

For the two-electrode measurement (full-cell), a Swagelok cell and a microfiber filter paper (separator) were used to assemble the symmetric device. All electrochemical tests were performed in 2.5 M KNO_3_ aqueous electrolyte at room temperature. 

The specific capacitance *C*_s_ (F g^−1^) of the half-cell was obtained from the GCD profiles using the following Equation [43]:(1)$$C_{s}=\frac{I\Delta t}{m\Delta V}$$
where *I* represents the current (mA), ∆*t* is the time (s) of the discharge slop from GCD, *m* is the mass (mg) of the active electrode and ∆*V* is the operating potential (V).

The specific capacitance *C*_s_ (F g^−1^), specific energy *E* (W h kg^−1^) and the specific power *P* (W kg^−1^) for the symmetric device were calculated using the mass total *m_T_* (mg) of the positive and negative electrode from the Equations (2)–(4) [44]:(2)$$C_{s}=\frac{I\Delta t}{m_{T}\Delta V}$$
(3)$$E=\frac{C_{s}\Delta V^{2}}{7.2}$$
(4)$$P=3600\frac{E_{d}}{\Delta t}$$

## 3. Results and Discussion

### 3.1. Structural, Morphological and Textural Characterization

XRD patterns of the as-prepared PAC/MoO_2_/Mo_2_C ternary composites are displayed in Figure 2a. These XRD spectra reveal diffraction peaks matching with the monoclinic MoO_2_ (ICSD card No. 86-0135, space group: P21/c, cell parameters: a = 5.6096 Å, b = 4.8570 Å, c = 5.6259 Å) and the hexagonal Mo_2_C (ICSD card No. 35-0787, space group: P63/mmc, with cell parameters: a = 3.0124 Å, b = 3.0124 Å, c = 4.7352 Å) in PAC/MoO_2_/Mo_2_C-0.5, PAC/MoO_2_/Mo_2_C-1 and PAC/MoO_2_/Mo_2_C-2 samples. All diffraction peaks located with approximate 2θ values of 26.1°, 37.1°, 53.7°, 60.7° and 66.9° corresponding to (011), (211), (220), (310) and (131) crystallographic planes, respectively can be assigned to the monoclinic MoO_2_ [45,46]. 

Therefore, the other featured peaks located at 2θ of 34.3°, 37.9°, 39.4°, 52.1°, 61.5°, 69.5°, 74.6° and 75.5° can be ascribed to the (100), (002), (101), (102), (110), (103), (112) and (201) crystallographic planes from the hexagonal Mo_2_C, respectively [47,48]. Additionally, the broad peak around 2θ of 26.5° corresponding to (002) diffraction of graphite (ICSD card No. 41-1487) could be attributed to the presence of the amorphous carbon domains of the PAC which overlaps with MoO_2_ peak at 26.1° [45,48].

The Raman spectra of the as-prepared PAC/MoO_2_/Mo_2_C porous ternary composites are shown in Figure 2b. The characteristic peaks at around 125, 152, 200, 342, 381 and 666 cm^−1^ bands could be attributed to the vibration modes of the monoclinic MoO_2_. The Raman active modes at 285, 825 and 998 cm^−1^ bands could be assigned to vibrational features of the Mo_2_C [49,50,51,52]. Two other characteristics peaks are also observed at (1341–1360 cm^−1^) and (1583–1609 cm^−1^) which correspond to the D and G bands, respectively (as shown in Table 1). The D band is associated to the disordered graphitic structure in carbon matrix while the G is due to the sp^2^-hybridized graphitic carbon [43]. The intensity ratio of D and G bands (I_D_/I_G_ ratio) recorded in Table 1 reveals the graphitization degree of the as-synthesized ternary composites [42]. The I_D_/I_G_ ratio values decreased from 1.03 to 0.97 with increasing mass ratio of the molybdenum precursor to the porous carbon. This indicates a balanced amorphous carbon to the graphitic carbon in the PAC/MoO_2_/Mo_2_C composites resulting from the MoO_2_ and Mo_2_C nanoparticles embedded into the porous carbon network [53,54].

High-resolution transmission electron microscopic (HRTEM) micrographs and selected area electron diffraction (SAED) patterns were further performed to provide more crystal structural information of the PAC/MoO_2_/Mo_2_C-0.5, PAC/MoO_2_/Mo_2_C-1 and PAC/MoO_2_/Mo_2_C-2 ternary composites as shown in Figure 3. Figure 3a–c revealed the HRTEM micrographs of the ternary composites in which the lattice fringes are highlighted in yellow arrow and the layer of amorphous PAC in orange arrow. The lattice fringes with an inter-planar spacing (d) approximate values of 0.340 nm, 0.219 nm and 0.283 nm are corresponding to the crystallographic planes (011), (−212) and (−102) of the monoclinic MoO_2_ (ICSD card No. 86-0135), respectively. The other d-spacing values of 0.237 nm and 0.227 nm are assigned to the crystallographic planes (002) and (101) of the hexagonal Mo_2_C (ICSD card No. 35-0787), respectively. The as-obtained SAED patterns of the PAC/MoO_2_/Mo_2_C ternary composites are exhibited in Figure 3d–f. The SAED patterns show the bright diffraction rings which are attributed to (302), (310) and (220) planes of MoO_2_ and those (002), (100), (101) and (201) planes to Mo_2_C. 

These results imply a successful incorporation of the MoO_2_ and Mo_2_C heterostructures into the amorphous PAC matrix through a pyrolysis process which are consistent with the XRD analysis in Figure 2a. 

The SEM micrographs of PAC/MoO_2_/Mo_2_C ternary composites prepared at different mass ratios of molybdenum precursor to PAC are displayed in Figure 4. The morphology of the ternary composites reveals the formation of a mixture of agglomerated nanoparticles and nanoplates embedded into the interconnected porous structure of the PAC at low magnification (Figure 4a,c,e).

An increase of molybdenum content reveals an increased tendency of both agglomerated nanoparticles and nanoplates morphologies as highlighted in circles in Figure 4b,d,f (high magnification). These two different morphologies are perhaps due to the presence MoO_2_ and Mo_2_C in the composites, but it is not easy to identify which of these belongs to a specific morphology. 

EDX mapping was also applied to determine the elemental distribution of the PAC/MoO_2_/Mo_2_C-0.5, PAC/MoO_2_/Mo_2_C-1 and PAC/MoO_2_/Mo_2_C-2 ternary composites as seen in Figure 5a–i. It is observed that the Mo, O and C elements are uniformly distributed throughout the interconnected porous carbon structure. This suggests that the agglomerated nanoparticles and nanoplates were composed of MoO_2_ and Mo_2_C embedded into the carbon matrix.

N_2_ adsorption/desorption experiment was conducted to investigate the textural properties of the PAC/MoO_2_/Mo_2_C ternary composites as shown in Figure 6 and Table 2. Figure 6a,b presents the sorption isotherms and the pore size distribution curves, respectively, of the ternary composites. All isotherms depicted a type IV features associated with a H4 hysteresis loop which indicates the coexistence of the micropores and mesopores structures in the ternary composites [55,56]. The BET specific surface area (SSA) values of PAC/MoO_2_/Mo_2_C-0.5, PAC/MoO_2_/Mo_2_C-1 and PAC/MoO_2_/Mo_2_C-2 samples are 804, 711 and 301 m^2^ g^−1^, respectively. A decrease in SSA was observed upon increasing the ammonium molybdate precursor loading from 0.5 to 2. For the total pore volume and micropore area (Table 2), the same trend is also identified for all samples from with the decrease from 0.44 to 0.20 cm^3^ g^−1^ and 670 to 165 m^2^ g^−1^, respectively.

The decrease in the SSA and total pore volume could be ascribed to the embedding of the MoO_2_ and Mo_2_C nanoparticles into the porous PAC during the pyrolysis process which can block some pores [57]. 

However, the SSA of the as-synthesized samples are much higher than that reported for similar materials such as MoO_2_/Mo_2_C/C composite prepared via ion-exchange method (73.4 m^2^ g^−1^) [58], MoO_2_/Mo_2_C/C spheres by hydrothermal and calcination processes (159.6 m^2^ g^−1^) [36] and MoO_2_/Mo_2_C/C microspheres obtained by a mild polymer regulation procedure followed by calcination treatment (57.6 m^2^ g^−1^) [59].

The formation of the nanoparticles MoO_2_ and Mo_2_C in the ternary composite emanates from the interaction between the ammonium molybdate tetrahydrate ((NH_4_)_6_Mo_7_O_24_·4H_2_O) and the porous PAC (denoted as C) at elevated temperature (≥800 °C) under argon atmosphere. It is good to mention that typically activated carbon (PAC) comprised of OH and COOH groups on the surface, which makes it acidic, favors a thermal reduction of the ammonium molybdate precursor to MoO_2_ instead of MoO_3_. The MoO_2_ could react with carbon at high temperature and under inert atmosphere (Ar) to give Mo_2_C. This process can be described with the following Equations [42,60]:(5)$${\left({\mathrm{NH}}_{4}\right)}_{6}{\mathrm{Mo}}_{7}O_{24}\;\cdot \;4H_{2}O\to {\mathrm{MoO}}_{2}+H_{2}O+{\mathrm{NH}}_{3}↑$$
(6)$$2C+4{\mathrm{MoO}}_{2}\to 2{\mathrm{Mo}}_{2}C+4O_{2}$$

During the pyrolysis, the ammonium molybdenum precursor decomposes to form MoO_2_, H_2_O and ammonia gas (NH_3_) being released at high temperature. In addition, the generated MoO_2_ nanoparticles could react with the porous carbon (PAC) leading to the formation of Mo_2_C. The formation of a ternary composite comprising of PAC, MoO_2_ and Mo_2_C could possibly enhance ion intercalation as well as create an interconnected porous network. This might promote an easy diffusion of the electrolyte’s ions through the electrode materials and further enhance the fast transport of the ions which are beneficial for the electrochemical analysis.

### 3.2. XPS Analysis 

The surface chemistry property and the elemental composition of the as-synthesized ternary composites were determined using X-ray photoelectron spectroscopy (XPS). The wide survey scan spectrum depicted the distinctive peaks of the carbon (C 1s), molybdenum (Mo3p_1/2_, Mo3p_3/2_ and Mo 3d) and oxygen (O 1s) elements in PAC/MoO_2_/Mo_2_C-0.5, PAC/MoO_2_/Mo_2_C-1 and PAC/MoO_2_/Mo_2_C-2 composites as illustrated in Figure 7a. 

Table 3 presents the atomic percentage (at.%) of C, Mo and O elements in the as-synthesized ternary composites. From these samples, it can be seen that the carbon content decreases from 73.9 to 54.9 at.% as the yield of the molybdenum increases. However, the molybdenum and oxygen contents were found to increase from 8.6 to 20.2 at.% and 17.5 to 24.9 at.%, respectively. Notably, the PAC/MoO_2_/Mo_2_C-2 material exhibited the smallest carbon content and highest molybdenum and oxygen contents. This could be due to the formation of MoO_2_ and Mo_2_C nanoparticles during the pyrolysis process. On the other hand, the elemental composition in the ternary composites is significantly influenced by the mass loading of molybdenum precursor into the PAC.

The high-resolution XPS spectra of the Mo 3d split into 3d_5/2_ and 3d_3/2_ spin-orbit components in the binding energy range of 226–241 eV as presented in Figure 7b–d. In Figure 7b, the deconvolution of the core level Mo 3d spectrum exhibits six sets of peaks which indicate the presence of four oxidation states Mo^2+^, Mo^4+^, Mo^5+^ and Mo^6+^ in PAC/MoO_2_/Mo_2_C-0.5 ternary composite. The peak located at 228.8 eV (Mo^2+^ 3d_5/2_) is associated to Mo-C bond in Mo_2_C while the pair of peaks located at 229.6 and 232.9 eV (Mo ^4+^ 3d_5/2_/3d_3/2_) are attributed to the formation of MoO_2_ [42,61,62,63]. The pair of peaks at binding energies of 231.0 and 234.4 eV (Mo^5+^ 3d_5/2_/3d_3/2_) and that located at 232.5 and 235.9 eV (Mo^6+^ 3d_5/2_/3d_3/2_) are the characteristics of the MoO_3_ which could be assigned to the surface oxidation and sample oxidation in air of the metastable phase of the MoO_2_ [64,65,66,67]. Figure 7c,d presents the fitting of the core level Mo3d in PAC/MoO_2_/Mo_2_C-1 and PAC/MoO_2_/Mo_2_C-2 ternary composites. In comparison with PAC/MoO_2_/Mo_2_C-0.5 ternary composite, there are no changes in the number of peaks deconvoluted which means that all the composites have similar oxidation states (Mo^2+^, Mo^4+^, Mo^5+^ and Mo^6+^). Appendix A presents the atomic percentage (at.%) of all deconvoluted peaks of the ternary composites. What is noticeable is that PAC/MoO_2_/Mo_2_C-1 has high at.% of Mo_2_C and MoO_2_ as compared to the other composites. 

This could be beneficial in the electrochemical measurements of these composites because these two materials are expected to improve the electrochemical properties of PAC where MoO_2_ is expected to contribute pseudocapacitive behavior, while Mo_2_C will contribute stability and conductivity. 

The high-resolution C 1s and O 1s core levels of the ternary composites are shown in Appendix A. The fitting of the C 1s spectrum (Appendix A) shows the characteristic peak of Mo-C bond in Mo_2_C at 283.3 ± 0.2 eV [68,69].

The other four peaks are attributed to the C=C (sp^2^ hybridized), C-C (sp^3^ hybridized), C-OH and O-C=O bonds corresponding to the binding energies at 284.4 ± 0.3eV, 285.2 ± 0.2 eV, 287.5 ± 0.6 and 290.7 ± 0.3 eV, respectively [70,71,72]. The deconvolution of the O 1s feature (Appendix A) exhibits a peak located at 530.6 ± 0.2 eV linked to Mo-O bond, the two other peaks at 533.3 ± 0.2 eV and 536.4 ± 0.2 eV are associated to C-O and O-C=O bonds, respectively [73,74,75]. The results of the XPS analysis confirm the formation of the MoO_2_ and Mo_2_C nanoparticles in all ternary composites which are consistent with the XRD, Raman, HRTEM and SAED results.

### 3.3. Electrochemical Characterization

All measurements of the ternary composite electrodes with different molybdenum precursor content were done first in a three-electrode configuration using 2.5 M KNO_3_ aqueous electrolyte. Figure 8a,b displays the comparative cyclic voltammetry (CV) profiles of the PAC/MoO_2_/Mo_2_C-0.5, PAC/MoO_2_/Mo_2_C-1 and PAC/MoO_2_/Mo_2_C-2 electrodes at a constant scan rate of 50 mV s^−1^ in both negative and positive potential windows of −0.9–0 V and 0–0.9 V vs. Ag/AgCl, respectively. A quasi-rectangular characteristic was observed for all CV profiles indicating an electrical double layer capacitor (EDLC) behavior for these samples [24]. It can be seen that the CV profile of PAC/MoO_2_/Mo_2_C-1 electrode displays a higher current response than other electrode materials. This superior current response could be assigned to the moderate loading of molybdenum precursor into the porous carbon network, which provided more active sites enhancing the fast ion diffusion and good interaction between the interface of electrode/KNO_3_ electrolyte. 

Appendix A display the galvanostatic charge-discharge (GCD) comparison curves of the PAC/MoO_2_/Mo_2_C ternary composites. 

The GCD curves of all electrodes are performed within both negative (−0.9–0.0 V vs. Ag/AgCl) and positive (0.0–0.9 V vs. Ag/AgCl) operating potential windows, respectively at a constant specific current of 1 A g^−1^. 

The GCD curves depicted a symmetrical triangular profile that confirms the electrical double layer capacitor nature of the ternary composites electrodes supported by the CV curves [76]. It is also observed that the GCD curve of the PAC/MoO_2_/Mo_2_C-1 electrode has a longer discharge time as compared to PAC/MoO_2_/Mo_2_C-0.5 and PAC/MoO_2_/Mo_2_C-2 electrodes agreeing with the CV results. Furthermore, the details of the CV curves at different scan rates from 10 to 100 mV s^−1^ and GCD profiles at specific currents ranging from 1 to 10 A g^−1^ of the PAC/MoO_2_/Mo_2_C-1 ternary composite because of superior electrochemical properties are shown in Appendix A, respectively. The corresponding specific capacitance (*C_s_*) as a function of the specific current in the range of 1–10 A g^−1^ is presented in Figure 8c,d in the negative (−0.9–0.0 V vs. Ag/AgCl) and positive (0.0–0.9 V vs. Ag/AgCl) operating potential windows for all three composites, respectively. The *C_s_* of the ternary composites was determined from the discharge period of GCD patterns using Equation (1). From both Figure 8c,d, it is observed that the PAC/MoO_2_/Mo_2_C-1 electrode depicted the highest *C_s_* value in both potential windows reflecting longer charge-discharge pattern which is consistent with the highest current response from the CV curve. However, the smallest *C_s_* value of PAC/MoO_2_/Mo_2_C-2 in both potential windows can be explained by the fact that further increasing the molybdenum precursor content could lead to the blockage of some pores in the porous carbon matrix as evidenced by a decrease in SSA of 301 m^2^ g^−1^ and thus limit the ion diffusion at the electrode/electrolyte interface [77].

Electrochemical impedance spectroscopy (EIS) analysis of the as-synthesized PAC/MoO_2_/Mo_2_C ternary composites was evaluated at open circuit voltage (*V*_OC_) in the frequency range of 100 kHz to 10 mHz. The EIS data are provided using Nyquist plot which illustrates the variation of the impedance as a function of the frequency as shown Figure 8e. 

The Nyquist plots of all ternary composites exhibit a semi-circle at high to medium frequency region corresponding to the charge transfer resistance (*R**_ct_*) and a quasi-straight line slightly tilted to *Z*” imaginary axis at low frequency region indicating the ion diffusion throughout the electrolyte [27,78]. The Nyquist curves of the ternary composites further confirms the capacitive characteristic. The intercept of the *Z*’ real axis (beginning of the arc) at high frequency depicts the equivalent series resistance (*ESR*) which represents the combination of resistance at electrode/electrolyte and electrode/current collector interfaces [79]. As seen in the inset to Figure 8e, PAC/MoO_2_/Mo_2_C-1 electrode depicts smaller *ESR* and *R**_ct_* values of 0.59 and 1.01 Ω as compared to PAC/MoO_2_/Mo_2_C-0.5 (0.76 and 1.10 Ω) and PAC/MoO_2_/Mo_2_C-2 electrodes (1.31 and 1.84 Ω), respectively. In addition, PAC/MoO_2_/Mo_2_C-1 electrode also has the shortest diffusion path length and closest to the *Z*” axis suggesting a quicker ion diffusion of the interfacial electrode and KNO_3_ electrolyte and shows better capacitive features among the ternary composites. 

Considering all the above results, the as-prepared PAC/MoO_2_/Mo_2_C-1 electrode recorded superior electrochemical performance among the other ternary composites. This might be ascribed to the synergistic effect of the MoO_2_, MO_2_C and porous carbon obtained after the reaction of ammonium molybdate and porous carbon at equal mass ratios during the pyrolysis route. This is also supported by the at.% of all the deconvoluted peaks in Appendix A where for this particular sample MoO_2_ and Mo_2_C show higher at.% as compared to the rest of the composites. The formation of the MoO_2_ and Mo_2_C nanoparticles embedded into the porous PAC provided abundant electro-actives sites for the charge transfer ability and quick ion diffusion of the electrolyte, which improve the wettability and the electrical conductivity, thus the charge storage of the electrode.

The electrochemical measurement of the PAC/MoO_2_/Mo_2_C-1 ternary composite electrode was further performed in a two-cell configuration by assembling a symmetric device using identical electrolyte. Figure 9a,b displays the CV features of the as-fabricated symmetric supercapacitor (PAC/MoO_2_/Mo_2_C-1//PAC/MoO_2_/Mo_2_C-1) under an operating cell potential of 0–1.8 V. The CV features of the ternary composite device reveals a quasi-rectangular behavior whereas the current response increased upon increasing the scan rates from 10 to 400 mV s^−1^ (Figure 9a) suggesting quasi-reversible electron transfer kinetics and dominated electrical double layer. 

A slight redox peak was observed in the CV curves due to the pseudocapacitive contribution from the molybdenum oxide. The minor redox reactions could emanate from the insertion of K^+^ ions into the MoO_2_ containing electrodes. A similar observation was reported by Wang et al. [80], on the lithium-ion insertion onto MoO_2_ that is associated with the monoclinic–orthorhombic–monoclinic phase transition of MoO_2_ [80]. As such a transition from MoO_2_ to K_X_MoO_2_ by the insertion of K^+^ ions to the molybdenum oxide can be postulated [81]. The CV curves of the PAC/MoO_2_/Mo_2_C-1//PAC/MoO_2_/Mo_2_C-1 device still maintain the rectangular-like feature upon increasing the scan rate to high rate from 0.5 to 2.5 V s^−1^ (as seen in Figure 9b) which demonstrated a high rate capability [82]. GCD plots of the assembled device in the operating potential of 0–1.8 V are shown in Figure 9c. A typical triangular behavior was recorded for all GCD plots at various specific currents from 1 to 20 A g^−1^ confirming the capacitive charge storage mechanism of the symmetric ternary composite device [24]. The plot of the obtained specific capacitance (*C*_s_) calculated using Equation (2) against the specific current from 1 to 20 A g^−1^ is depicted in Figure 9d. The recorded value *C_s_* of the ternary composite device was found to be 115 F g^−1^ at 1 A g^−1^ specific current. The ternary composite PAC/MoO_2_/Mo_2_C-1//PAC/MoO_2_/Mo_2_C-1 device still delivered a high *C_s_* of 67 F g^−1^ even after a twentyfold increase of specific current which confirms the good rate capability of 58.3% obtained from the symmetric supercapacitor.

Figure 10a depicts the specific energy against specific power (Ragone plot) measured at various specific currents (1–20 A g^−1^). The symmetric ternary composite device recorded high specific energy of 51.8 W h kg^−1^ with an associated power of 0.9 kW kg^−1^ at 1 A g^−1^. Interestingly, the symmetric ternary composite device can maintain up to 30 W h kg^−1^ of specific energy with a corresponding specific power of 18 kW kg^−1^ even at 20 A g^−1^ increase of specific current. These specific energy/power values recorded for the PAC/MoO_2_/Mo_2_C-1//PAC/MoO_2_/Mo_2_C-1 symmetric cell are better than reports on Mo-based/C composites for supercapacitors applications as shown in Appendix A.

To investigate the stability of the PAC/MoO_2_/Mo_2_C-1//PAC/MoO_2_/Mo_2_C-1 device, the cycling test based on the long-term galvanostatic charge-discharge was performed at 10 A g^−1^ specific current as shown in Figure 10b. 

The ternary composite device recorded a columbic efficiency of 99.8% up to 25,000 charge-discharge cycles and a capacitance retention found to be 94%, 92% and 83% after 7000, 15,000 and 25,000 constant charge-discharge cycles, respectively. These results indicate that even up to 25,000 continuous cycling the ternary composite device still maintains good stability with a specific capacitance loss of 17% as compared to the initial value. The good long-term stability of the symmetric device is owed to the rapid electron transfer kinetics offered by the ternary composite electrode. 

An additional stability performance, floating test (or voltage holding) has been investigated on the PAC/MoO_2_/Mo_2_C-1//PAC/MoO_2_/Mo_2_C-1 symmetric supercapacitor. Figure 10c displays the variation of the specific capacitance versus the floating time of each 10 h during 150 h at a maximum potential cell of 1.8 V at 10 A g^−1^. An increase of 21% from the initial value of the specific capacitance is observed during the first 60 h of floating subsequently stabilizing up to 150 h floating time. The specific capacitance of the ternary composite device was enhanced from 74.7 to 90 F g^−1^ after the floating time which also highlights an improvement of the specific energy from 33.7 to 40.2 W h kg^−1^. The improvement of the PAC/MoO_2_/Mo_2_C-1//PAC/MoO_2_/Mo_2_C-1 device in the specific capacitance and specific energy could be ascribed to more penetration of ions electrolyte into the network of MoO_2_ and Mo_2_C nanoparticles embedded in the porous carbon. This could consequently increase the electrode wettability and enable faster diffusion of electrolyte ions at the electrode/KNO_3_ electrolyte interface, hence enhancing the charge storage [83,84]. In brief, the as-fabricated PAC/MoO_2_/Mo_2_C-1//PAC/MoO_2_/Mo_2_C-1 device revealed a good stability performance in terms of long-term cycling up 25,000 cycles and floating time over 150 h thereby implying a superior electrochemical performance of the entire device.

Figure 10d illustrates the Nyquist plots of PAC/MoO_2_/Mo_2_C-1//PAC/MoO_2_/Mo_2_C-1 symmetric supercapacitor before and both after long-term cycling and holding test. All Nyquist plots exhibit a nearly vertical feature at low frequency referring to the ideal capacitive characteristic and the great electrical conductivity of the PAC/MoO_2_/Mo_2_C-1 ternary composite. A slight decrease of the *ESR* values was observed from the original value of 0.82 Ω to 0.78 Ω and 0.74 Ω for both after 25,000 cycles and 150 h floating test respectively as shown in the inset to Figure 10d. Similarly, the *R_ct_* value depicted also a small decrease from the initial value of 1.14 Ω to 1.11 Ω and 1.04 Ω after cycling and holding tests, respectively. These small *ESR* and *R_ct_* values demonstrate a fast charge transport capability and rapid ion diffusion through the full symmetric device which got improved by the stability tests as indication that the electrode had better wettability after stability [85]. 

The Bode plot of the as-fabricated-symmetric device which defines the plot of the angle phase versus frequency is shown in Figure 10e before and after cycling stability and voltage holding. The phase angle values increased from −78° to −80° and −85° after cycling and holding test, respectively. These values are close to −90° which confirm the ideal capacitive behavior of the PAC/MoO_2_/Mo_2_C-1 ternary composite [24]. 

According to these results, the as-fabricated symmetric supercapacitor demonstrated a superior electrochemical performance after cycling and holding test in aqueous electrolyte which could be due to the fact that the electrodes wettability had been improved and hence ions have more access to the pores. The high performance of the PAC/MoO_2_/Mo_2_C-1//PAC/MoO_2_/Mo_2_C-1 symmetric supercapacitor is based on the synergistic effect of the ternary composite with the following benefits: (i) high electrical conductivity of Mo_2_C, (ii) pseudo capacitor effect of MoO_2_ and (iii) large surface area of porous carbon (PAC). Owing to these advantages, the PAC/MoO_2_/Mo_2_C-1//PAC/MoO_2_/Mo_2_C-1 can be used as an excellent charge storage device. 

## 4. Conclusions

A ternary peanut shell activated carbon/molybdenum oxide/molybdenum carbide (PAC/MoO_2_/Mo_2_C) composite was successfully synthesized via a facile in-situ pyrolysis route of ratio of porous carbon to different mass loading of ammonium molybdate (1:0.5; 1:1; 1:2). All as-synthesized materials display the in-situ formation of MoO_2_ and Mo_2_C nanostructures into the porous carbon based on the XRD, Raman, HRTEM, SAED, EDX mapping and XPS analysis. The ternary composite with the mass ratio of 1 to 1 (PAC/MoO_2_/Mo_2_C-1) provided a superior electrochemical characteristic in a neutral 2.5 M KNO_3_ electrolyte. The as-assembled PAC/MoO_2_/Mo_2_C-1//PAC/MoO_2_/Mo_2_C-1 symmetric device delivered an excellent specific capacitance of 115 F g^−1^ at 1 A g^−1^ with a good rate capability (58% at 20 A g^−1^) and cycling stability (99.8% columbic efficiency after 25,000 cycles). Moreover, a specific energy of 51.8 W h kg^−1^ with a corresponding power of 0.9 kW kg^−1^ was recorded for the symmetric device within an operating potential window of 1.8 V at 1 A g^−1^ specific current. Interestingly, the electrochemical results of the device show a significant enhancement of 21% from its initial specific capacitance value after 160 h holding test. These remarkable performances are linked to the great synergistic effect of the different components into the ternary composite by supplying favorable properties: Pseudo capacitor behavior of the MoO_2_, highly conductive Mo_2_C and high surface area of the porous carbon. This study provides a simple and low-cost way to enhance the electrochemical performance of carbons by incorporating Mo-based components into porous activated carbon.

## Figures and Tables

**Figure 1 nanomaterials-11-01056-f001:**
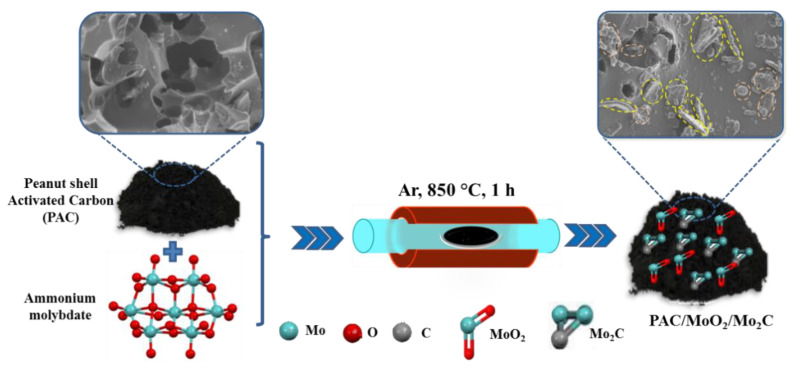
Schematic illustration for the synthesis method of the PAC/MoO_2_/Mo_2_C ternary composites.

**Figure 2 nanomaterials-11-01056-f002:**
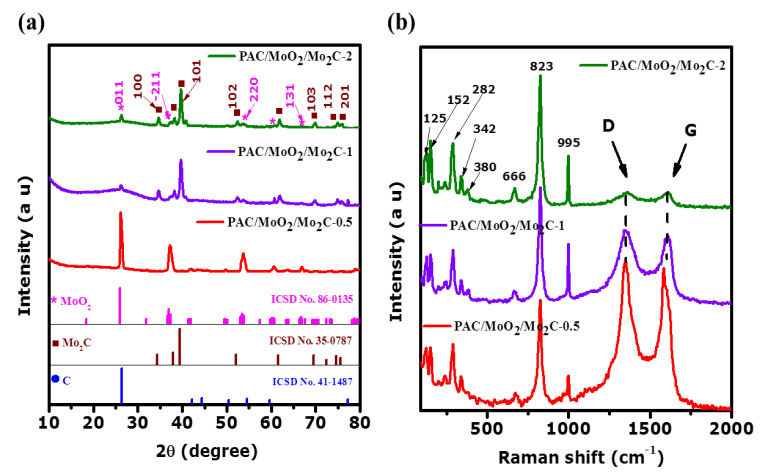
(**a**) XRD patterns and (**b**) Raman spectra of the PAC/MoO_2_/Mo_2_C-0.5, PAC/MoO_2_/Mo_2_C-1 and PAC/MoO_2_/Mo_2_C-2.

**Figure 3 nanomaterials-11-01056-f003:**
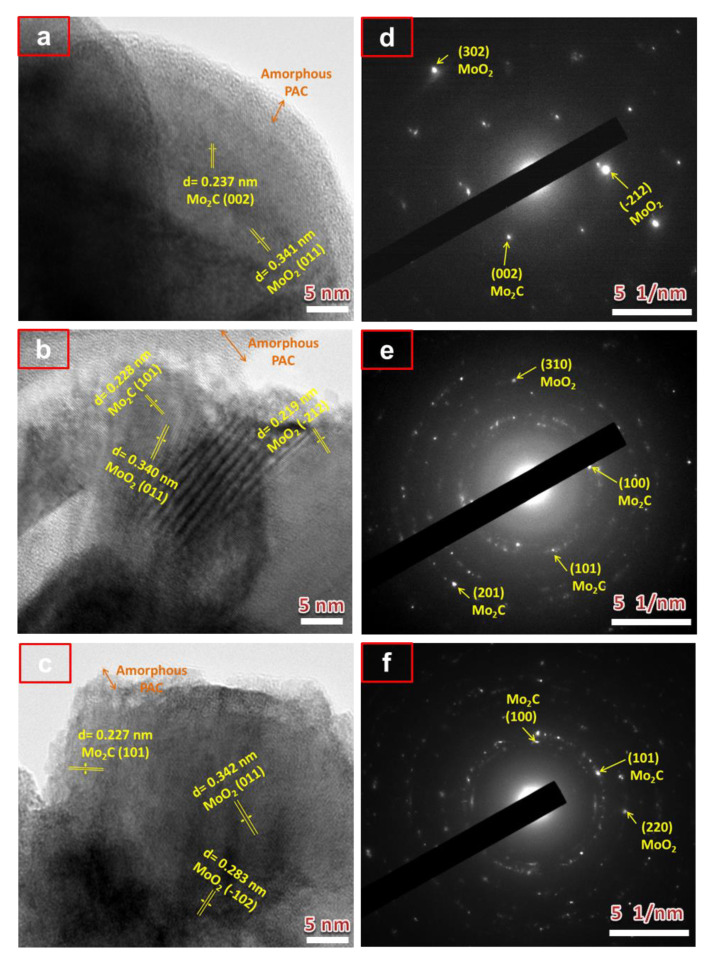
HRTEM micrographs and SAED patterns of (**a**,**d**) PAC/MoO_2_/Mo_2_C-0.5, (**b**,**e**) PAC/MoO_2_/Mo_2_C-1 and (**c**,**f**) PAC/MoO_2_/Mo_2_C-2 ternary composites, respectively.

**Figure 4 nanomaterials-11-01056-f004:**
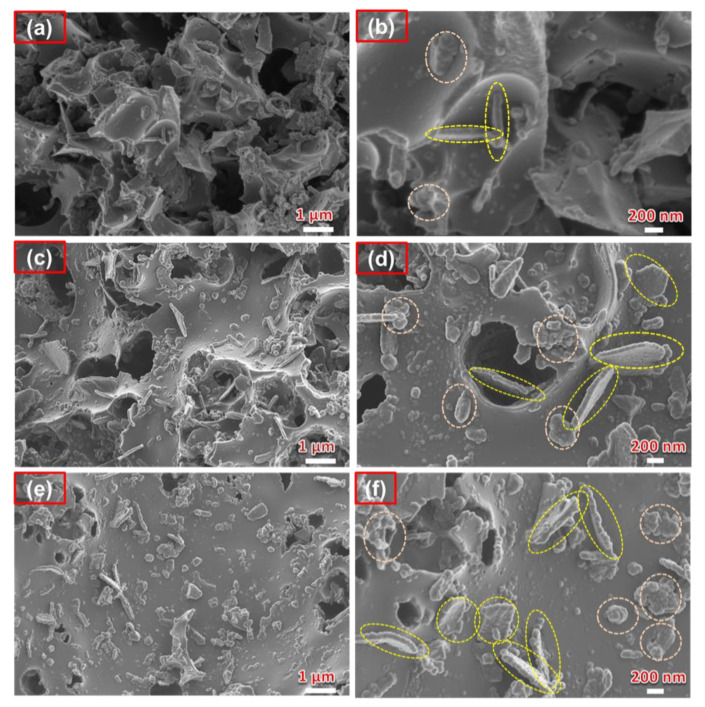
SEM micrographs at low and high magnification of (**a**,**b**) PAC/MoO_2_/Mo_2_C-0.5, (**c**,**d**) PAC/MoO_2_/Mo_2_C-1 and (**e**,**f**) PAC/MoO_2_/Mo_2_C-2 ternary composites.

**Figure 5 nanomaterials-11-01056-f005:**
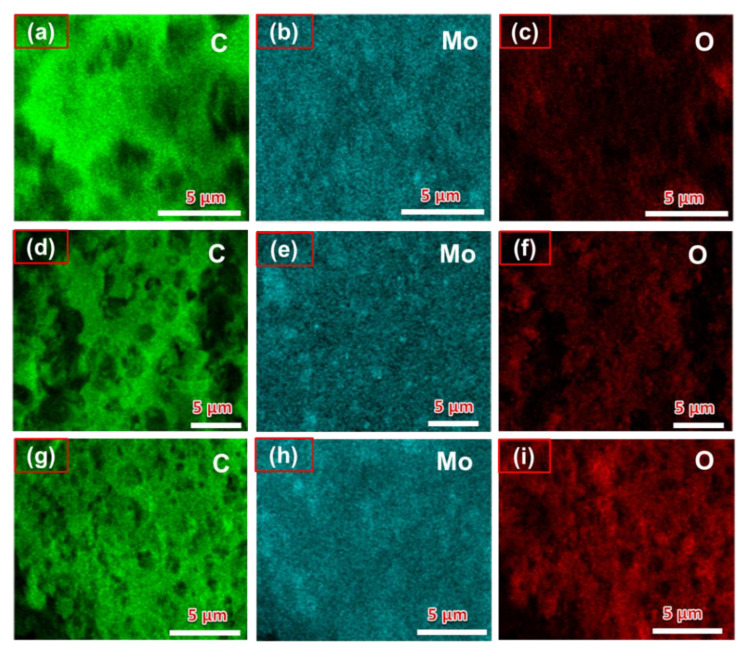
EDX mapping images showing the distribution of C, Mo and O individual elements in: (**a**–**c**) PAC/MoO_2_/Mo_2_C-0.5, (**d**–**f**) PAC/MoO_2_/Mo_2_C-1 and (**g**–**i**) PAC/MoO_2_/Mo_2_C-2 ternary composites.

**Figure 6 nanomaterials-11-01056-f006:**
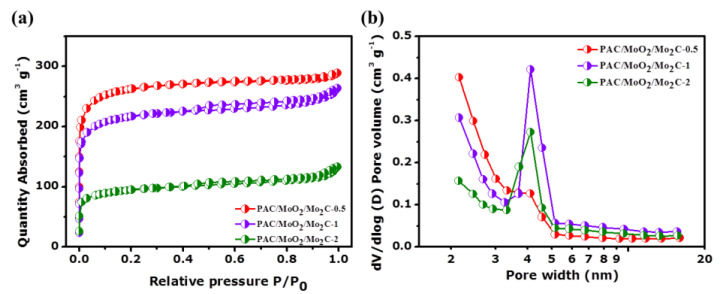
(**a**) N_2_ absorption-desorption isotherms, (**b**) pore size distribution of the PAC/MoO_2_/Mo_2_C-0.5, PAC/MoO_2_/Mo_2_C-1 and PAC/MoO_2_/Mo_2_C-2.

**Figure 7 nanomaterials-11-01056-f007:**
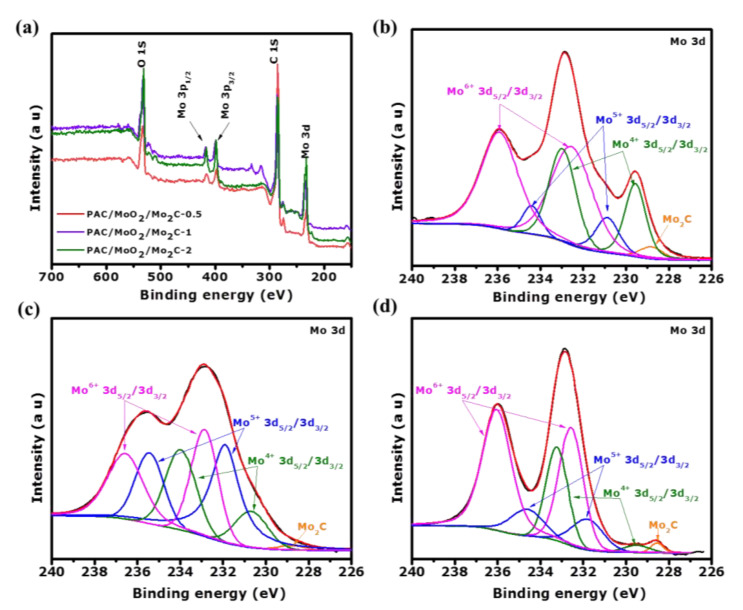
(**a**) XPS survey spectra of the as-synthesized ternary composites and high resolution of Mo 3d, (**b**) PAC/MoO_2_/Mo_2_C-0.5, (**c**) PAC/MoO_2_/Mo_2_C-1 and (**d**) PAC/MoO_2_/Mo_2_C-2 ternary composites.

**Figure 8 nanomaterials-11-01056-f008:**
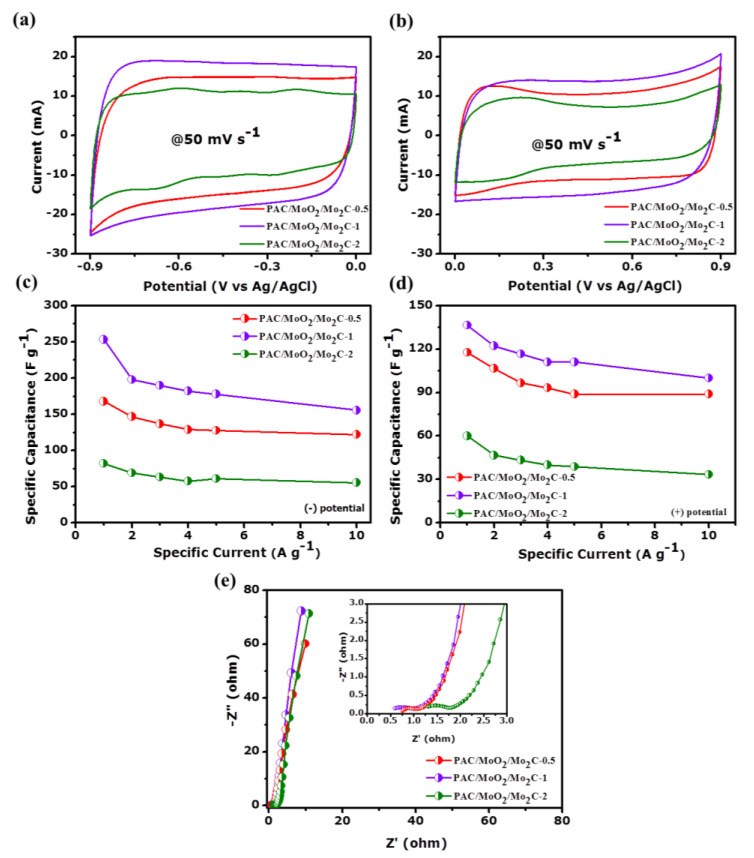
(**a**,**b**) Cyclic voltammetry at scan rate of 50 mV s^−1^, (**c**,**d**) specific capacitance at different specific current values in (−0.9–0.0 V) and (0.0–0.9 V) operating potential windows and (**e**) Nyquist plots of the PAC/MoO_2_/Mo_2_C ternary composites in a three-electrode configuration.

**Figure 9 nanomaterials-11-01056-f009:**
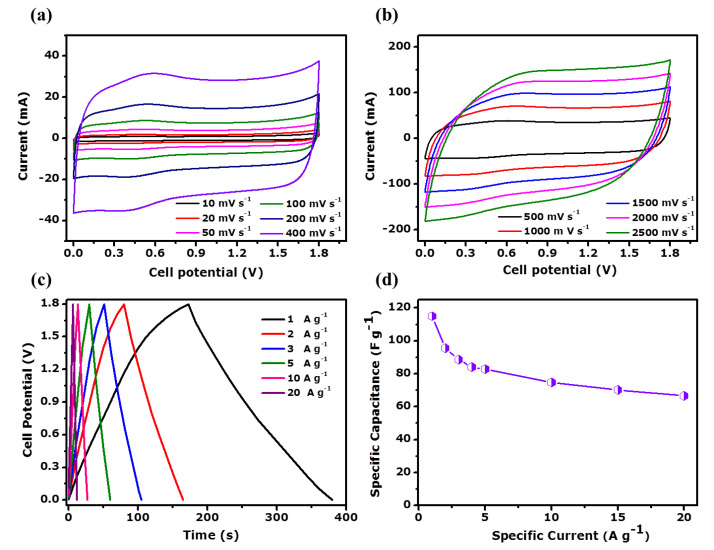
(**a**,**b**) Cyclic voltammetry at scan rates variant from 10–400 mV s^−1^ and from 500–2500 mV s^−1^, respectively, (**c**) galvanostatic charge-discharge profiles at various specific currents from 1 to 20 A g^−1^ and (**d**) specific capacitance as a function of specific current for PAC/MoO_2_/Mo_2_C-1//PAC/MoO_2_/Mo_2_C-1 symmetric cell.

**Figure 10 nanomaterials-11-01056-f010:**
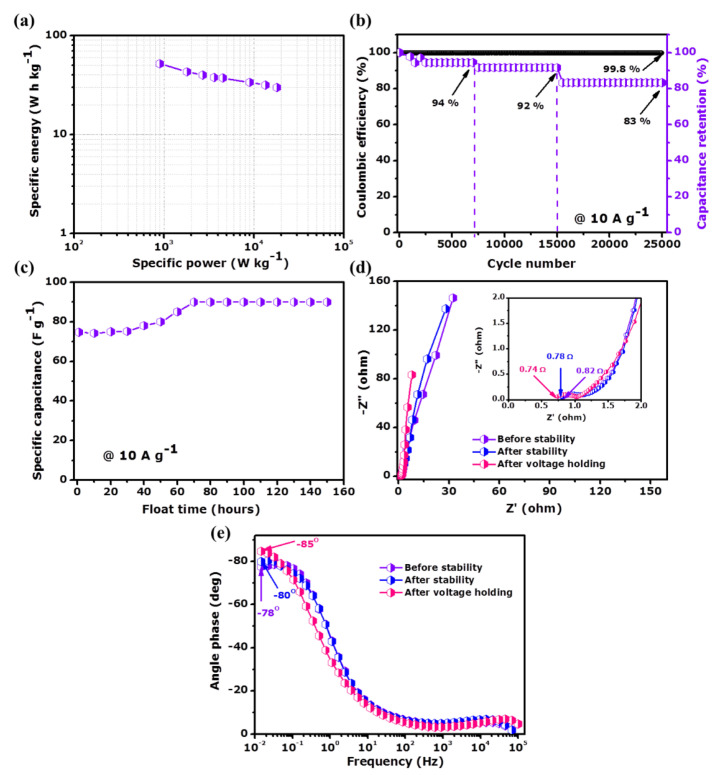
(**a**) Ragone plot, (**b**) cycling stability up to 25,000 cycles, (**c**) specific capacitance as function of voltage holding time up to 180 h, (**d**) Nyquist plots and (**e**) Bode plot of PAC/MoO_2_/Mo_2_C-1//PAC/MoO_2_/Mo_2_C-1 symmetric cell.

**Table 1 nanomaterials-11-01056-t001:** Raman data of the PAC/MoO_2_/Mo_2_C ternary composites.

Samples	D-Band (cm^−1^)	G-Band (cm^−1^)	I_D_/I_G_ Ratio
PAC/MoO_2_/Mo_2_C-0.5	1349	1583	1.03
PAC/MoO_2_/Mo_2_C-1	1341	1607	1.01
PAC/MoO_2_/Mo_2_C-2	1360	1610	0.97

**Table 2 nanomaterials-11-01056-t002:** Textural properties of the PAC/MoO_2_/Mo_2_C ternary composites.

Samples	BET SSA (m^2^ g^−1^)	Total Pore Volume (cm^3^ g^−1^)	Micropore Volume (cm^3^ g^−1^)	Micropore SSA (m^2^ g^−1^)	Mesopore Volume (cm^3^ g^−1^)
PAC/MoO_2_/Mo_2_C-0.5	804	0.44	0.23	670	0.21
PAC/MoO_2_/Mo_2_C-1	711	0.40	0.19	575	0.21
PAC/MoO_2_/Mo_2_C-2	301	0.20	0.07	165	0.13

**Table 3 nanomaterials-11-01056-t003:** Elemental composite of the PAC/MoO_2_/Mo_2_C ternary composites.

	Elemental Composition (at.%)		
Samples	C 1s	O 1s	Mo 3d
PAC/MoO_2_/Mo_2_C-0.5	73.9	17.5	8.6
PAC/MoO_2_/Mo_2_C-1	62.4	22.6	15.0
PAC/MoO_2_/Mo_2_C-2	54.9	24.9	20.2

## Data Availability

Data can be available upon request from the authors.

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
