# Peer review of "Enhanced Electrochemical Behavior of Peanut-Shell Activated Carbon/Molybdenum Oxide/Molybdenum Carbide Ternary Composites"

_nanomaterials, 2021, doi:10.3390/nano11041056_

Round 1

Reviewer 1 Report

Summary:

The manuscript nanomaterials-1176000 titled, “Enhanced electrochemical behaviour of peanut-shell activated carbon/molybdenum oxide/molybdenum carbide ternary composites,” report the use of biomass-waste activated carbon/molybdenum oxide/molybdenum carbide ternary composites as the electrode material in super capacitor. The concept and experimental design are interesting. The study gives fruitful experiment and analysis to support the concept and the assumption. The manuscript is well-organized and gives good reading experience.

General comment:

This is an interesting work showing a new concept in active material preparation. The detailed material, electrochemical, and supercapacitor analysis are very good. A minor revision is therefore suggested.

Comments:

(1) Minor redox reactions are found, and might belong to MoO2 and Mo2C. The reaction and mechanism are suggested to be discussed during the revision.

Please revise manuscript by explaining the redox reaction that might be contributed by MoO2 and Mo2C by additional electrochemical analysis to reference papers.

(2) It is impressive that 105 papers are cited here. It is not necessary to do so for a research article. The readers would not focus on the reference citation, but they would focus on the results and discussion. In addition, the initial 52 references are not mentioned again in the experiment and result and discussion section. This implies reference 53-105 are related to this work, while references 1-52 not.

Please revise manuscript by refining the references.

(3) The Y-axis shown in Figure 6 is suggested to be [cm3 nm-1 g-1] to show the effect of micropore and mesopore. It is also suggested to calculate the micropore area and mesopore volume and area in Table 2.

Please revise manuscript by analyzing the pore volume and surface area of micropores and mesopores.

(4) What is the conductivity of the sample? Is 850C good enough to cook the sample?

Please conducting conductivity analysis on the samples.

Author Response

Reviewer 1

(1) Minor redox reactions are found, and might belong to MoO2 and Mo2C. The reaction and mechanism are suggested to be discussed during the revision.

Please revise manuscript by explaining the redox reaction that might be contributed by MoO2 and Mo2C by additional electrochemical analysis to reference papers.

Response 1:

 We thank the reviewer for raising this invaluable comment concerning the reaction mechanism. The following statement has been added in the manuscript to address this concern:

“The minor redox reactions could emanate from the insertion of K+ ions into the MoO2 containing electrodes. A similar observation was reported by Wang et al. [80], on the lithium-ion insertion onto MoO2 that is associated with the monoclinic–orthorhombic–monoclinic phase transition of MoO2 [80]. As such a transition from MoO2 to KXMoO2 by the insertion of K+ ions to the molybdenum oxide can be postulated [81]”

(2) It is impressive that 105 papers are cited here. It is not necessary to do so for a research article. The readers would not focus on the reference citation, but they would focus on the results and discussion. In addition, the initial 52 references are not mentioned again in the experiment and result and discussion section. This implies reference 53-105 are related to this work, while references 1-52 not.

Please revise manuscript by refining the references.

Response 2:

Authors appreciate the reviewer for raising this issue. Authors have streamed down the total number of references. It wouldn’t make sense to get rid of all the references in the introduction section even if they don’t appear in the results discussion because some of these references gives basis of the use of these materials in other energy storage and conversion.

(3) The Y-axis shown in Figure 6 is suggested to be [cm3 nm-1 g-1] to show the effect of micropore and mesopore. It is also suggested to calculate the micropore area and mesopore volume and area in Table 2.

 Please revise manuscript by analyzing the pore volume and surface area of micropores and mesopores

Response 3:

We thank the reviewer in this regard. The correct units for the Y-axis shown in Figure 6 are as already shown [cm3 g-1] not [cm3nm-1g-1) as it refers to the quantity absorbed per unit mass.

We have upgraded all the parameters that can be obtained from the TriStar II 3020 system software in the table 2 in the revised manuscript as shown below:

 Table 2. Textural properties of the PAC/MoO2/Mo2C ternary composites

Samples

BET SSA

(m2 g-1)

Total pore volume (cm3 g-1)

Micropore volume (cm3 g-1)

Mesopore volume

(cm3 g-1)

 Micropores SSA (m2 g-1)

PAC/MoO2/Mo2C-0.5

804

0.44

0.23

0.21

670

PAC/MoO2/Mo2C-1

   711

0.40

0.19

0.21

575

PAC/MoO2/Mo2C-2

  301

0.20

0.07

0.13

165

(4) What is the conductivity of the sample? Is 850 ºC good enough to cook the sample?

Please conducting conductivity analysis on the samples.

 Response 4:

We appreciate the reviewer’s comment, but since our samples are in powder form it wasn’t easy to conduct the electrical conductivity measurements. However equivalent series resistance (ESR) was extrapolated to evaluate the electronic conductivity of the electrode materials as shown in Fig.8e. Further explanations on the ESR are highlighted in the revised manuscript.

Yes indeed 850 ºC is enough to obtain the ternary composite materials. This has already been established in our previous study as per the following references:

1. Manyala, A. Bello, F. Barzegar, A.A. Khaleed, D.Y. Momodu, J.K. Dangbegnon, Coniferous pine biomass: A novel insight into sustainable carbon materials for supercapacitors electrode, Materials Chemistry and Physics, 182, 2016, 139-147. doi.org/10.1016/j.matchemphys.2016.07.015.

2. Ndeye F. Sylla, Ndeye M. Ndiaye, Balla D. Ngom, Bridget K. Mutuma, Damilola Momodu, Mohamed Chaker, Ncholu Manyala, Ex-situ nitrogen-doped porous carbons as electrode materials for high performance supercapacitor, Journal of Colloid and Interface Science, 569, 2020, 332-345, doi.org/10.1016/j.jcis.2020.02.061.

Reviewer 2 Report

In this paper, authors report MoO2/Mo2C in activated carbon nanostructure for supercapacitor applications, which can be extended to other energy conversion/storage technologies as well. Minor revision is suggested as the data is complete, but reference list needs improvement.

What are some limitations of this work and how can it be improved? Pls discuss.

Introduction talks about importance of renewable energy and sustainable future, pls cite these papers: 10.1126/science.aad4998 and 10.1038/nmat4834.

Interfaces in supercapacitors are very important and widely discussed in this work, please cite the following papers: 10.1038/s41578-021-00303-1 and 10.1126/science.1249625.

Overall a great paper.

Author Response

Reviewer 2

  1. In this paper, authors report MoO2/Mo2C in activated carbon nanostructure for supercapacitor applications, which can be extended to other energy conversion/storage technologies as well. Minor revision is suggested as the data is complete, but reference list needs improvement.

 Response 1:

The reference list has been improved.

  1. What are some limitations of this work and how can it be improved? Pls discuss.

Response 2:

As we have already mentioned in our introduction section that this ternary composite has been reported before in the battery application (please refer to references 36 and 41 in the manuscript). We have also indicated that to the best of our knowledge it is the 1st time that this ternary composite is applied as electrodes for supercapacitors. There has been reports where MoO2/C and Mo2/C composites have been applied as the electrodes for supercapacitors individually and each have its own advantages as mentioned in the introduction part. But the combination of these compounds have proved to be more beneficial, and hence one cannot see much limitations.

  1. Introduction talks about importance of renewable energy and sustainable future, pls cite these papers: 10.1126/science.aad4998 and 10.1038/nmat4834.

Interfaces in supercapacitors are very important and widely discussed in this work, please cite the following papers: 10.1038/s41578-021-00303-1 and 10.1126/science.1249625.

Response 3:

The authors have added the papers and highlighted them in the revised manuscript.